# Study on the Decoupling and Interaction Effect between Industrial Structure Upgrading and Carbon Emissions under Dual Carbon Targets

**DOI:** 10.3390/ijerph20031945

**Published:** 2023-01-20

**Authors:** Yuqing Sun, Yingchao Liu, Zhiwei Yang, Mengyao Wang, Chunying Zhang, Liya Wang

**Affiliations:** 1College of Economics, North China University of Science and Technology, Tangshan 063210, China; 2College of Artificial Intelligence, North China University of Science and Technology, Tangshan 063210, China; 3College of Science, North China University of Science and Technology, Tangshan 063210, China

**Keywords:** industrial structure upgrading, carbon emissions, decoupling model, nuclear density function, Dagum Gini coefficient

## Abstract

The issue of climate and environment has been paid more and more attention by countries all over the world, especially regarding carbon emissions. Many national policies and scholars’ research contents have focused on this issue, which has become a hot topic in today’s society. As the world’s largest carbon emitter, it is vital for China to achieve green development, upgrade its industrial structure and explore the relationship between industrial structure upgrading and carbon emissions. To explore the decoupling and interactive effects of industrial structure upgrading and carbon emissions, this paper divides industrial structure upgrading into two aspects: rationalization of industrial structure and upgrading of industrial structure. Indicators related to industrial structure upgrading and carbon emissions are selected and the decoupling model of carbon emissions and industrial structure upgrading is constructed using panel data from 30 regions from 1997 to 2019. The core density function is used to analyze the decoupling distribution characteristics, and then the Gini coefficient decomposition method is used to analyze the carbon emissions decoupling index, revealing the regional differences and sources of carbon emissions decoupling index. Finally, spatial factors are included in the regression model to verify the spatial synergy effect of industrial structure upgrading on carbon emissions. The overall and local Moran indexes are used to reveal the spatial internal structure and agglomeration characteristics of industrial structure upgrading and carbon emissions, and, based on the research results, policy recommendations are put forward to promote sustainable and stable development of industrial structure upgrading in China. This provides a new perspective for understanding the relationship between industrial structure upgrading and carbon emissions and also provides a decision-making reference for promoting decoupling of industrial structure upgrading and carbon emissions under high-quality economic development and forcing low-carbon transformation of the industrial structure.

## 1. Introduction

Global warming is an urgent problem to be solved in the world today, which is related to human survival and sustainable development. It not only generates various extreme weather phenomena but has adverse effects on natural ecosystems [1]. Relevant data show that China’s carbon dioxide emissions in 2021 will be 11.9 billion tons, accounting for 33% of the global total, and China’s carbon emissions will rank first in the world. As the country with the largest carbon emissions in the world, striving to achieve a carbon peak by 2030 and carbon neutrality by 2060 (referred to as the “double carbon” goal) is an important strategic decision for China to address the global climate problem. However, most of China’s industries have low added value and are still in the “double low” stage of development. The resulting crisis of increasing total carbon emissions is still prominent. To reduce carbon emissions, cope with global warming and adapt to the requirements of high-quality development of China’s economy, China has continuously stressed the vital role of industrial structure upgrading in achieving the “double carbon” goal and put forward clear requirements in the five-year plan. The report of the 19th CPC National Congress pointed out that the main driving force of China’s carbon emissions reduction lies in scientific and technological innovation and industrial structure upgrading. As one of the three major methods of energy conservation and emission reduction, upgrading the industrial structure plays an important role in achieving the goal of carbon intensity and is also a requirement for responding to economic transformation and achieving high-quality economic development. In recent years, China’s industrial structure has been continuously upgraded and has become an essential part of the global value chain. Especially since the dual carbon target was put forward in September 2020, it is very important to achieve green development, upgrade the industrial structure and explore the relationship between industrial structure upgrading and carbon emissions.

(1) Research on the measurement and influencing factors of industrial structure upgrading: Gou Limin [2] applied the Lilien index model to measure the speed of industrial structure transformation to upgrade in 30 provinces in China and found that the level of urbanization and scientific and technological investment has a significant positive impact on the speed of industrial structure transformation and upgrading. To measure upgrading of industrial structure, Wang et al. [3] introduced the hierarchical coefficient of industrial structure to explain the upgrading level of industrial structure of provinces, regions and cities and empirically analyzed the impact of digital economy on upgrading of industrial structure in China. Guo et al. [4] studied the impact of digital HP finance on upgrading of industrial structure and selected the advanced industrial structure index as the proxy variable for upgrading of regional industrial structure to calculate. Liu et al. [5] used the entropy weight method to calculate the weight of industrial structure upgrading and rationalization indicators and weighted the two indicators to calculate the industrial structure upgrading level index. Chen et al. [6] measured the upgrading of industrial structure from three dimensions, namely upgrading of the industrial structure, rationalization of industrial structure and low carbonization of industrial structure, and then synthesized the three indicators through entropy method. Wang et al. [7] constructed the industrial structure upgrading and rationalization index to measure industrial structure upgrading.

(2) Research on carbon emission measurement and influencing factors: Cui Heng [8] and Chen et al. [9] used energy consumption and energy carbon emissions coefficient to calculate carbon emissions and determined that the influencing factors of carbon emissions include economic growth, industrial structure, energy structure, etc. Zhang et al. [10] used the carbon emissions coefficient method to measure the agricultural carbon emissions, structure and carbon emission efficiency in Guizhou Province and analyzed the sensitive relationship between agricultural carbon emissions and economic development through Tapio elasticity index. Yang et al. [11] measured the total carbon emissions by energy consumption and conversion coefficient of energy to standard coal and carbon emissions coefficient. Yang et al. [12] believe that China’s carbon emissions mainly come from consumption of fossil energy and use energy consumption, standard coal conversion coefficient, energy carbon emissions coefficient and carbon multiplier factor to calculate the carbon dioxide emissions. Xie et al. [13] used the natural logarithm of the sum of carbon dioxide emissions as the proxy indicator of carbon emissions.

(3) Research on the relationship between industrial structure upgrading and carbon emissions: On the one hand is the impact of industrial structure upgrading on carbon emissions. Some scholars believe that upgrading of industrial structure promotes carbon emissions reduction. Yang Xiaoxi, Wu L, et al. [12,14] believe that upgrading of the industrial structure would promote technological innovation, improve the energy consumption structure, increase research, development and use of clean energy and reduce carbon emissions. Xie Wenqian, Li W, et al. [13,15] believe that upgrading of the industrial structure is conducive to technological progress and can develop new low-carbon technologies to reduce carbon emissions in the production process. Some scholars also believe that upgrading the industrial structure may inhibit carbon emissions. Gong et al. [16] found that adjustment of the industrial structure to pollution-intensive industries will inevitably lead to an increase in carbon dioxide emissions. If the industrial structure is adjusted to clean industries, it will reduce carbon dioxide emissions. On the other hand is the impact of carbon emissions on upgrading the industrial structure. Wang et al. [17] concluded that increasing carbon emissions has a significant inhibitory effect on upgrading and rationalization of the industrial structure. Dong B, Zheng J, et al. [18,19] believe that carbon emissions reduction could promote upgrading of the industrial structure.

According to the relevant literature at home and abroad, relevant research under the “double carbon” policy has increasingly become the focus of scholars. The research of scholars has deepened the understanding of upgrading the industrial structure and carbon emissions, but the existing research is mainly at the national or provincial level, ignoring the current situation of carbon emissions, the upgrading process of the industrial structure and the resource endowment between the east, middle and west of China. Additionally, some problems still need to be solved: what is the decoupling relationship between industrial structure upgrading and carbon emissions under the dual carbon target nationwide? What about the interaction effect? What are the spatial metrological characteristics? There are few studies that address these problems. In view of this, based on the era background of “dual carbon,” this paper measures the types of decoupling between industrial structure upgrading and carbon emissions in China’s provinces and analyzes the heterogeneity of decoupling types in the east, middle and west with the help of nuclear density function and Dagum Gini coefficient. Then, spatial measurement is used to explore the interaction effect between carbon emissions and transformation and upgrading of the industrial structure so as to reveal the internal relationship between industrial structure upgrading and carbon emissions and provide a policy basis for realization of China’s “dual carbon” goal, which has important practical significance for promoting green, low-carbon and high-quality economic development. Due to China’s large geographical area, the process and interaction of industrial structure upgrading and carbon emissions decoupling in each province are all different, so this paper divides China into eight major economic regions, which is helpful to analyze the differences between regions and the demonstration and guidance for each region and build different measurement models based on panel data to explore the heterogeneity and spatial spillover effects of carbon emissions and industrial structure upgrading decoupling so as to be more comprehensive and accurately grasp the effect between carbon emissions and industrial structure upgrading.

The main structure of this paper is as follows. The Section 2 introduces the decoupling model, kernel density function, Dagum Gini coefficient and Moran index and establishes an index system. The Section 3 shows the decoupling model of industrial structure upgrading and carbon emissions and analyzes heterogeneity with the help of the kernel density function and Gini coefficient. The Section 4 establishes the Dubin space model to analyze the synergy between industrial structure and carbon emissions. The Section 5 is the summary of this paper.

## 2. Research Design

### 2.1. Construction of Indicator System

Through query and analysis of relevant literature and official data, it is found that there are many indicators related to upgrading of the industrial structure and carbon emissions, including total carbon dioxide emissions, energy structure, technological innovation, environmental planning, urbanization level, population density, etc. In consideration of the comprehensiveness and accuracy of the subsequent model establishment and analysis, according to the existing literature, this paper finally selected eight indicators: total carbon dioxide emissions, total GDP, advanced industrial structure, rationalization of industrial structure, energy structure, energy intensity, degree of government intervention, R&D investment and Gini index. Table 1 shows the construction of industrial structure upgrading and carbon emissions indicator system.

#### 2.1.1. Measurement of Industrial Structure Upgrading

Upgrading of the industrial structure includes two definitions: one is the degree of coupling and coordination of production factors between different industrial sectors, that is, rationalization of the industrial structure. The second is the evolution process of the industrial structure to a high-level structure, that is, upgrading of the industrial structure. Therefore, this paper measures upgrading of the industrial structure from two dimensions: upgrading of the industrial structure and rationalization of the industrial structure.

(1)Advanced industrial structure (*SH*)

This paper measures the upgrading process of the industrial structure by the ratio of the gross domestic product of the tertiary industry to the gross domestic product of the secondary industry [20]. The higher the SH value is, the more service-oriented the economic structure is.

(2)Rational structure of production (*TL*)

This paper uses the Thiel index to measure the level of industrial structure rationalization [21]:(1)$$TL=-\sum_{i=1}^{3}(\frac{Y_{i}}{Y})\ln(\frac{Y_{i}}{L_{i}}/\frac{Y}{L})$$

In the formula, $Y_{i}$ and $L_{i}$ represent the GDP value (CNY 100 million) and employment number (10,000 people) of the ith industry in each province, respectively. The above formula provides the deviation degree of the average industrial structure. In addition, to overcome the defect that indicators such as the Thiel index and the degree of deviation of industrial structure are “inverse indicators” to measure the level of the industrial structure rationalization, a negative sign is added to the front of the whole to turn them into positive indicators to measure the level of industrial structure rationalization. Therefore, this paper uses this indicator to measure the level of industrial structure rationalization. The closer TL is to 0, the more reasonable the industrial structure is; on the contrary, the industrial structure is unreasonable.

(3)Industrial structure upgrading index (*IS*)

The weighted average method is used to measure the upgrading of industrial structure from the two indicators of industrial structure upgrading and industrial structure rationalization, namely:(2)$$IS=\frac{SH+TL}{2}$$

#### 2.1.2. Measurement of Carbon Emissions

Carbon emissions are mainly generated from combustion of fossil fuels. Therefore, this paper uses the existing research and selects the consumption of eight fossil fuels, such as diesel, coke, coal, kerosene, gasoline, fuel oil, natural gas and crude oil, which are closely related to carbon emissions according to the carbon emissions coefficient published by IPCC to estimate carbon emissions. The estimation formula is as follows:(3)$$CI=\sum_{i=1}^{H}E_{i}\times CEFI_{i}$$
(4)$$CEFI_{i}=H_{i}\times CH_{i}\times COR_{i}\times CEF_{i}\times \frac{44}{12}\times 10^{-6}$$
where $E_{i}$ is the ith energy consumption, $CEFI_{i}$ is the carbon emissions coefficient of energy i, $H_{i}$ is the average low calorific value, $CH_{i}$ is the carbon emission factor, $COR_{i}$ is the carbon oxidation rate and $CEF_{i}$ is the carbon emissions coefficient.

### 2.2. Research Method

#### 2.2.1. Decoupling Model

“Decoupling” indicates that the interaction between multiple economic systems has weakened or disappeared and has become the main way to analyze resource and environmental issues in economics. To explain the uncertainty of decoupling status of dynamic data and the accuracy of decoupling prediction, incremental data are introduced into the Tapio index model, the decoupling status is defined by the range of elastic value and the changing relationship between carbon emissions and industrial structure upgrading is reflected by the Tapio decoupling index, thus forming a new decoupling model that describes the relationship between carbon emissions change and industrial structure upgrading. The formula is as follows:(5)$$\varphi_{CI,IS}=\frac{\Delta CI/CI}{\Delta IS/IS}$$

The decoupling states are divided into eight categories according to the positive and negative values of ΔCI/CI and ΔIS/IS and the value of decoupling elasticity, as shown in Table 2. Where, + represents the change rate of carbon emissions or industrial structure upgrading index greater than 0, − represents less than 0.

#### 2.2.2. Kernel Density Function

As one of the nonparametric estimation methods, the kernel density function estimation method can effectively test the dynamic evolution trend of the sample distribution. Specifically, the kernel density function estimation method uses a smooth peak function to match the sample data. This method estimates the probability density of random variables and converts the data of random variables into a continuous density curve, which can reflect the distribution position, shape, extensibility and polarization of random variables. On this basis, the dynamic evolution trend of decoupling index in the eastern, central and western regions of China can be obtained. The empirical distribution function is:(6)$$F_{n}\left(y\right)=\frac{1}{n}\sum_{i=1}^{n}I\left(x_{i}\leq y\right)$$
where n represents the number of observations. I(z) is the conditional relation. When z is true, I(z)=1; when z is false, I(z)=0; therefore, this method assumes that the density function of the random variable X is as follows:$$f\left(x\right)=\left[F_{n}\left(x+h\right)-F_{n}\left(x-h\right)\right]$$
$$=\frac{1}{2h}\left(\frac{1}{n}\sum_{i=1}^{n}I\left(x-h\right)\leq x_{i}\leq x+h\right)=\frac{1}{nh}\sum_{i=1}^{n}\left(\frac{1}{2}I\left(-1\leq \frac{x-x_{i}}{h}\leq 1\right)\right)$$
(7)$$=\frac{1}{nh}\sum_{i=1}^{n}K\left[(x_{i}-\overset{-}{x})/h\right]$$
where $x_{i}$ is the independent identically distributed observation value, $\overset{-}{x}$ is the average value, K is the nuclear density and h is the bandwidth. Generally, smaller bandwidth is selected to ensure higher accuracy of $x_{i}$. As a weighted number or smooth conversion function, the kernel density generally needs to meet the following conditions:(8)$$K\left(x\right)\geq 0,K\left(x\right)=K\left(-x\right),\int_{-\infty }^{+\infty }K\left(x\right)dx=1,\sup K\left(x\right)<+\infty ,\int_{-\infty }^{+\infty }K^{2}\left(x\right)dx<+\infty$$

The kernel function is mainly used for weighted or smooth conversion. The form of the kernel function is as follows:(9)$$f\left(x\right)=\frac{1}{\sqrt{2\pi }}\exp\left(-\frac{x^{2}}{2}\right)$$

#### 2.2.3. Dagum Gini Coefficient

In this paper, Dagum Gini coefficient and its sub-group decomposition method are used to analyze the regional gap of the decoupling index of carbon emissions and industrial structure upgrading. Compared with the traditional Gini coefficient and Theil index, the Dagum Gini coefficient and subgroup decomposition method can effectively avoid the problems of sub-sample distribution, cross overlap and regional difference.

According to the similarity and interregional differences of natural and economic and social conditions in the economic zone, the country is divided into eight economic zones, as shown in Table 3:

Dagum Gini coefficient calculation formula is as follows:(10)$$G=\frac{\sum_{j=1}^{k}\sum_{h=1}^{k}\sum_{r=1}^{n_{j}}\sum_{r=1}^{n_{h}}\left|y_{ji}-y_{hr}\right|}{2n^{2}\overset{-}{y}}$$
where k is the total number of provinces in the region, 8 in this paper, i and r represent the serial number of provinces in the region, $n_{j}$ and $n_{h}$ represent the number of provinces in j and h, respectively, y represents the decoupling index, n represents the total number of provinces, 30 in this paper, and $\overset{-}{y}$ represents the average value of the decoupling index.

The idea of Dagum Gini coefficient decomposition is to divide the total Gini coefficient into intraregional difference $G_{w}$, inter regional difference $G_{nb}$ and hypervariable density $G_{t}$. The relationship between the three is $G=G_{w}+G_{nb}+G_{t}$. The calculation formula is:(11)$$G_{jj}=\frac{\frac{1}{2y}\sum_{i=1}^{n_{j}}\sum_{r=1}^{n_{j}}\left|y_{ji}-y_{hr}\right|}{n_{j}^{2}}$$
(12)$$G_{w}=\sum_{j=1}^{k}G_{jj}p_{j}s_{j}$$
(13)$$G_{jh}=\frac{\sum_{i=1}^{k}\sum_{r=1}^{n_{h}}\left|y_{ji}-y_{hr}\right|}{n_{j}n_{h}\left(\overset{-}{y_{j}}-\overset{-}{y_{h}}\right)}$$
(14)$$G_{nb}=\sum_{j=2}^{k}\sum_{h=1}^{j-1}G_{hj}\left(p_{j}s_{h}+p_{h}s_{j}\right)D_{jh}$$
(15)$$G_{t}=G_{nb}=\sum_{j=2}^{k}\sum_{h=1}^{j-1}G_{jh}\left(p_{j}s_{h}+p_{h}s_{j}\right)\left(1-D_{jh}\right)$$

The above formula, $p_{j}=\frac{n_{j}}{n}$, $s_{j}=\frac{n_{j}\overset{-}{y_{j}}}{n\overset{-}{y}}$; $D_{jh}$ refers to the relative regional influence between economic zone j and economic zone h in the eight comprehensive economic zones. $d_{jh}$ is the difference of the decoupling index, and $p_{jh}$ is the first order of overshoot. The calculation formula is as follows. $F_{j}$ and $F_{h}$ are the cumulative density distribution functions of economic zone j and economic zone h.
(16)$$D_{jh}=\frac{d_{jh}-p_{jh}}{d_{jh}+p_{jh}}$$
(17)$$d_{jh}=\int_{0}^{\infty }dF_{j}\left(y\right)\int_{0}^{y}\left(y-x\right)dF_{h}\left(x\right)$$
(18)$$p_{jh}=\int_{0}^{\infty }dF_{h}\left(y\right)\int_{0}^{y}\left(y-x\right)dF_{j}\left(x\right)$$

#### 2.2.4. Moran Index

In the field of spatial econometrics, the Moran index is widely used to measure spatial correlation. Moran index can be divided into global or local spatial autocorrelation Moran index. The global spatial autocorrelation Moran index reflects the similarity of carbon emissions of neighboring provinces or municipalities, and its calculation formula is:(19)$$Moran\prime s=I=\frac{\sum_{i=1}^{n}\sum_{j=1}^{n}W_{ij}\left(Y_{i}-\bar{Y}\right)\left(Y_{j}-\bar{Y}\right)}{S^{2}\sum_{i=1}^{n}\sum_{j=1}^{n}W_{ij}}$$

Moran’s *I* is between 1 and −1. When the value is greater than 0, it indicates that the carbon emission of each province is positively correlated in space. The closer to 1, the greater the concentration of carbon emissions between provinces and cities. When it is equal to 0, it means that there is no spatial aggregation, that is, random aggregation. When it is less than 0, it means that the degree of carbon emissions among provinces is negatively correlated in space.

### 2.3. Data Source and Processing

In view of the availability of data, the relevant data of 30 provinces (cities, autonomous regions) in China from 1997 to 2019 were selected for the study (Tibet, Hong Kong, Macao and Taiwan, China were excluded in consideration of the availability of data. Due to the impact of COVID-19 epidemic, China’s economic growth has fluctuated significantly since 2020, which is an outlier. Therefore, to exclude the impact of outliers on the regression results, the sample period ended in 2019.). The data used are mainly from official websites, such as the National Bureau of Statistics, provincial statistical offices, statistical yearbooks, etc. The data are more authoritative and accurate.

Due to the uneven original data, low-quality data may reduce the accuracy of the model, so the data need to be preprocessed before model establishment. In this paper, Python is used to process the missing value of the original data by linear regression fitting. In addition, natural logarithms are taken for each variable to eliminate heteroscedasticity as much as possible.

## 3. Analysis of the Heterogeneity of Decoupling between Industrial Structure and Carbon Emissions

### 3.1. Analysis of Decoupling Measurement Results

Considering the impact of time factors on the decoupling results, this paper further divides the sample periods into five periods according to the five-year plan: 1997–2000, 2001–2005, 2006–2010, 2011–2015 and 2016–2019. The model results are shown in Figure 1. Among them, there is no data in Tibet.

It can be seen from Figure 1 that, between 1997 and 2019, the decoupling index and decoupling status of all provinces in China changed greatly.

From 1997 to 2000, most regions of China were in a strong negative decoupling state. The decoupling index of industrial structure upgrading and carbon emissions in most provinces was less than 0, and the change rate of carbon emissions was positive, while the industrial structure upgrading index was negative, indicating that carbon emissions were on the rise, while the industrial structure upgrading index was on the decline, showing reverse growth, and the carbon emission rate was far greater than the growth rate of the industrial structure upgrading index. It shows that the deterioration of China’s ecological environment during this period is far faster than the upgrading of industrial structure. From 2001 to 2005, there were only two decoupling states in China: expansive negative decoupling state and strong negative decoupling state, indicating that, with the advancement of industrialization and urbanization, the demand for energy in various regions is also increasing, leading to an increase in carbon emissions, and the upgrading index of industrial structure is also gradually increasing. From 2006 to 2010, in addition to the expansive negative decoupling state and the strong negative decoupling state, there was expansive coupling, indicating that the carbon emissions and the industrial structure upgrading index increased simultaneously. In general, the decoupling state changed little in the same period. From 2011 to 2015, most regions of China were in a strong decoupling state and a weak decoupling state. From the perspective of change trend, the transition from a strong negative decoupling state to a strong decoupling state showed that, under the correct guidance of the five-year plan, carbon emissions gradually decreased and the industrial structure upgrading index gradually increased. The decoupling index of industrial structure upgrading and carbon emissions has declined year by year, which shows that the dependence of carbon emissions on industrial structure upgrading is also steadily declining. From 2016 to 2019, 14 provinces were in weak decoupling; that is, China’s carbon emissions growth rate was slower and slower compared with the industrial structure upgrading index, indicating that the country’s green and low-carbon transformation and development has achieved great results.

### 3.2. Analysis of Decoupling Distribution Characteristics Based on Kernel Density Function

To further characterize the dynamic evolution trend of carbon emissions decoupling index, this paper chooses the kernel density estimation method for further analysis. At the same time, to enhance the diversity of analysis angles and the pertinence of research conclusions, 30 provinces and cities are divided into three regions, namely the east, the middle and the west, with reference to the regional standard of the National Bureau of Statistics, and the comparative research results are shown in the following Table 4:

### 3.3. Analysis of Difference Sources Based on Dagum Gini Coefficient Decomposition Method

#### 3.3.1. Overall Differences

To calculate the overall difference, intraregional difference, net contribution of inter-regional difference and hypervariable density between regions of carbon emissions decoupling index, Dagum Gini coefficient decomposition method is adopted in this paper to reveal the regional difference size and source of carbon emissions decoupling index. The results are shown in Figure 2.

As shown in Figure 2, the development of China’s carbon emissions decoupling index is relatively stable. Between 1997 and 2019, the overall Gini coefficient of the country was between 0.146 and 0.181. Specifically, from 1997 to 2004, it was relatively stable and fluctuated slightly. By 2005, it had a relatively high rise. From 2005 to 2017, it had stable development. By 2018, it had a relatively high rise for the second time. In general, the change was small, and the overall difference of decoupling index was increasing.

#### 3.3.2. Differences within the Region

The difference between the three regions in China from 1997 to 2019 is very small, with a small upward trend. The reason for this phenomenon may be that the decoupling index of each province is declining with continuous promotion of regional low-carbon ecological construction, thus increasing the regional difference. As can be seen from Figure 3, the Gini coefficient curve in each region shows evolution characteristic of “stable fluctuation”. Through horizontal comparison, except for the middle reaches of the Yellow River, the Gini coefficient of other regions is still lower than the overall national level.

Figure 3 shows the regional differences in the carbon emissions decoupling index. Specifically, the Gini coefficient in the middle reaches of the Yellow River is much higher than that in other economic regions, and it has a continuous upward trend, which indicates that the regional development in the middle reaches of the Yellow River is increasingly unbalanced. The main reason for the rebound of the Gini coefficient in 2017 is that the Gini coefficient in the middle reaches of the Yellow River fluctuates greatly. From the perspective of regional differences, the top three annual average Gini coefficients of the eight economic zones are the middle reaches of the Yellow River, the middle reaches of the Yangtze River and the southern coastal areas. Except for the southwest region, the internal differences of other regions are gradually increasing, with an average annual decline of 0.961%.

#### 3.3.3. Differences between Regions

It can be seen from Figure 2 that the differences between regions generally fluctuate in a regular manner, rising first and then falling. Overall, the regional difference value shows an upward trend, and the difference gradually increases slowly, but the difference change is still small, indicating that the region should actively implement green and low-carbon measures to further promote coordinated development between regions.

### 3.4. Source and Contribution of Difference

From Figure 4, we can find that the collinearity between regional differences and the carbon emissions decoupling index is the largest, ranging from 0.547 to 0.664. During the study period, the differences between regions fluctuated but generally showed an increasing trend. The contribution rate of hypervariable density takes second place, ranging from 0.269 to 0.379. The contribution rate has little change during the investigation period, showing a downward trend overall. The contribution rate of regional difference is the smallest, between 0.066 and 0.075, and the contribution rate is stable.

To sum up, the average annual growth rate of the contribution rate of intraregional, interregional and hypervariable density to the carbon emissions decoupling index is −7.56%, 11.4% and −15.0%, respectively, which shows that there is a significant difference between upgrading of the industrial structure and the carbon emissions decoupling index in the region, and the regional low-carbon ecological development is unbalanced. In addition, the influence of interregional development on the development of the carbon emissions decoupling index is increasing and the contribution rate of intraregional and super-variable density to volatility is decreasing, which means that the impact of the overlapping problem of intraregional and super-variable density on the carbon emissions decoupling index is gradually decreasing.

## 4. Study on the Synergistic Effect of Industrial Structure and Carbon Emissions

### 4.1. Spatial Autocorrelation Analysis and Results

#### 4.1.1. Global Moran Index

First, this paper uses the global Moran index to test the spatial autocorrelation of the industrial structure upgrading index and carbon emissions in the sample period. The results are shown in Table 5. It is assumed that the spatial relationship between the observation results is constant during the study period. The results show that, although the industrial structure upgrading index and Moran’s *I* index of carbon emissions fluctuate, they are both positive at a significant level of 5%, indicating that the industrial structure upgrading index and carbon emissions have significant spatial positive correlation characteristics, specifically aggregation of high-level provinces, and the spatial dependence is relatively stable. Therefore, in general, spatial correlation plays a significant role, and it is appropriate to choose a spatial econometric model.

#### 4.1.2. Local Moran Index

For space reasons, only the results of 1997 and 2019 are reported. It can be seen from Figure 5 and Figure 6 that the corresponding nodes of the industrial structure upgrading indicators and the Moran index of carbon emissions among regions are mainly distributed in the first and third quadrants; that is, the regions have a strong directional promotion effect in local space, which is consistent with the test conclusion of the global Moran index. From 1997 to 2019, the total number of regions falling between the first quadrant and the third quadrant increased, which shows that the positive correlation between the industrial structure upgrading index and carbon emissions in some regions has increased. Therefore, due to the apparent positive correlation characteristics of local space, the influence of spatial factors should be fully considered to optimize the measurement mode of space.

### 4.2. Impact of Industrial Structure Upgrading on Carbon Emissions

#### 4.2.1. Impact of China’s Overall Industrial Structure Upgrading on Carbon Emissions

The industrial structure upgrading index and carbon emissions have a significant spatial correlation. By establishing a spatial measurement model, the impact of the industrial structure upgrading index on carbon emissions can be more accurately measured.

(1)LM inspection

To select an appropriate spatial panel model, this paper conducts a spatial correlation test on ordinary static panel regression (OLS), including LM Lag and stable LM Lag tests and LM Error and stable LM Error tests. As shown in Table 6, LM Error, the stable LM Error test and the stable LM Lag test all reject the original hypothesis, indicating that the samples selected in this paper have dual effects of spatial lag and spatial error autocorrelation. Since the spatial Dubin model SDM model can consider multiple effects at the same time, it can be preliminarily judged that selection of SDM is reasonable.

(2)Model comparison

According to the results of the Hausman test, SAR and SEM models select random effects [22], and the SDM model, Hausman, passes the test at 1% significance level and selects a fixed effect model. In selection of spatial metrological models, the SDM model has the smallest $\sigma^{2}$ and the largest goodness of fit $R^{2}$, so the fitting degree of the SDM model is the best. In conclusion, the fixed effect model of SDM is the most appropriate one. Stata15.1 software is used for all models, and the specific results are shown in Table 7.

(3)LR inspection

Finally, it is necessary to conduct an LR test on the spatial Dubin fixed effect model to verify whether the spatial Dubin model will degenerate into the spatial autoregressive model and spatial error model. According to Table 6, the index values of the LR test are 36.16 and 14.85, respectively, and the spatial autoregression model rejects the original hypothesis at the significance level of 1%. To sum up, the fixed effect model of spatial Dubin can be selected to study the impact of industrial structure upgrading on carbon emissions.

#### 4.2.2. Impact of Industrial Structure Upgrading on Carbon Emissions in Eastern, Central and Western Regions

Considering the influence of regional heterogeneity, the analysis is further divided into eastern, central and western regions. The results are shown in Table 8 and Table 9.

The Hausman results in Table 6 show that the fixed effect model of SDM should be selected. The LR test index values of the SAR model and SEM model in the eastern region are 25.820 and 30.010, respectively, and both reject the original hypothesis at the significance level of 1%. The LR test index values of the SAR model and SEM model in the central region were 43.250 and 39.440, respectively, which rejected the original hypothesis at the 1% significance level. The LR test index values of the SAR model and SEM model in the western region were 12.820 and 10.040, respectively, and the spatial autoregression model rejected the original hypothesis at the 10% significance level.

#### 4.2.3. Analysis of Regression Results of Spatial Econometric Model

According to the results of the fixed effect model of the SDM in the overall and eastern, central and western regions, the coefficient of carbon emissions is positive in China as a whole and in the central region, and the overall coefficient of China passes the hypothesis test at the 10% significance level, indicating that, the higher the industrial structure upgrading index of each province, the higher the carbon emissions of each province. However, the coefficient of carbon emissions is negative in the eastern region, and, through hypothesis testing at the 1% significance level, it shows that, the higher the industrial structure upgrading index of each province in the eastern region is, the lower the carbon emissions of each province, indicating that the industrial structure level in the eastern region is higher and it is easier to achieve the “dual carbon” goal. The carbon emissions coefficient in the central and western regions is not significant, indicating that the carbon emissions in this region do not have a significant spatial effect.

Because the spatial economy Dubin model can be understood as the spatial economic relationship between provinces, its parameter estimation results do not directly reflect the direct impact and spatial spillover impact. Therefore, this paper divides the various variables of society as a whole and their impact coefficients on the industrial structure upgrading index into direct effects, indirect effects and comprehensive effects.

As shown in Table 10, the direct impact of carbon emissions is positive but has not passed the significance test. Both the indirect effect and total effect are negative and have passed the 1% significance test, indicating that the carbon emissions of each province have a significant spatial impact on the upgrading of the industrial structure. If the interaction of spatial factors is ignored, the impact of industrial structure upgrading on carbon emissions will be underestimated. Therefore, it is reasonable to choose a spatial measurement model again.

## 5. Conclusions and Policy Implications

### 5.1. Conclusions

At present, most of the existing studies on industrial structure upgrading and carbon emissions are at the national or provincial level, and there is no explanation for links between regions within China. Compared with other existing studies, this paper shifts the research level to the eastern, central and western regions of China, revealing differences in the decoupling index of industrial structure upgrading and carbon emissions and evolution of the spatial pattern between the regions in China. This paper uses rationalization and upgrading to measure the industrial structure upgrading index, establishes an indicator system and uses panel data of 30 regions from 1997 to 2019 to build a decoupling model of carbon emissions and industrial structure upgrading and uses the nuclear density function and Dagum Gini coefficient decomposition method to analyze the distribution of carbon emissions decoupling index and regional differences and sources. Finally, spatial factors are included in the regression model to verify the spatial synergy effect of industrial structure upgrading on carbon emissions, and the global and local Moran index is used to reveal the spatial internal structure and agglomeration characteristics of industrial structure upgrading and carbon emissions. The main conclusions are as follows:

First, between 1997 and 2019, the upgrading of China’s industrial structure and carbon emissions have an unstable decoupling relationship, which generally follows the law of strong negative decoupling–strong decoupling–weak decoupling. The growth rate of carbon emissions is slower and slower compared with the upgrading index of the industrial structure. China has made progress in promoting low-carbon development.

Second, there is spatial heterogeneity between industrial structure upgrading and carbon emissions. From the national nuclear density function distribution, China’s carbon emissions decoupling index shows an upward trend. From the perspective of shape, the distribution of China’s carbon emissions decoupling index shows a trend of multi-polarization, and the distribution of carbon emissions decoupling in different provinces has certain differences. From the peak, there is a large gap between the distribution of carbon emissions decoupling in various provinces, and the phenomenon of two-level differentiation has become weak from strong.

Third, the Dagum Gini coefficient decomposition method is used to calculate the overall difference, regional difference, net contribution of regional difference and hypervariable density between the regions of the carbon emissions decoupling index. The upgrading of the industrial structure within the region is significantly different from the carbon emissions decoupling index, and the regional low-carbon ecological development is unbalanced. The impact of regional development on the development of the carbon emissions decoupling index is increasing, and the impact of the overlapping problem of regional and hypervariable density on the carbon emissions decoupling index is gradually decreasing.

Fourth, the industrial structure upgrading index and carbon emissions have a significant positive spatial correlation, and the carbon emissions of each province have a significant spatial spillover effect on the industrial structure upgrading. Among the control variables, the growth rate of economic development level is lower than that of industrial structure upgrading, thus hindering upgrading of the industrial structure. Influenced by regional policies, the “crowding out” effect of resources is obvious, which is shown as negative spillover; the development level of science and technology still needs to be improved; improvement in residents’ income level and urbanization level promote upgrading of the industrial structure.

### 5.2. Policy Implications

To achieve high-quality economic development and the “double carbon” goal, China needs to rely on upgrading of the industrial structure. To this end, the following recommendations are made:

First, optimize upgrading of the industrial structure and advocate low-carbon life. Promote transformation of industries to green and low carbon, eliminate backward production methods, upgrade and rationalize the industrial structure at the same time, optimize and upgrade traditional industries by using science and technology and advanced development models, promote energy conservation and consumption reduction and, finally, adjust and optimize the industrial structure and reduce carbon sources, as well as improve environmental awareness, advocate low-carbon life and jointly promote the process of upgrading the industrial structure and decoupling carbon emissions.

Second, promote regional cooperation and coordinated development. The study found that the industrial structure upgrading index and carbon emissions have significant positive spatial correlation characteristics, and the regional differences between industrial structure upgrading and carbon emissions decoupling index in the eastern, central and western regions have an increasing trend. Strengthen interregional contacts and exchanges and form a regional coordinated governance pattern through the radiative driving effect and demonstrate a leading role between cities. Strengthen inter-provincial technical exchange and cross-regional cooperation to promote coordinated control of carbon dioxide emissions among regions and promote upgrading of the industrial structure.

Third, combine regional characteristics and adjust measures to local conditions. There are obvious differences in the effects of industrial structure upgrading and carbon emissions. The higher the industrial structure upgrading index in the eastern region, the lower the carbon emissions, while the opposite is true in the central and western regions. Therefore, for the eastern region, it is necessary to raise standards, resist high-emission enterprises and lay the foundation for carbon neutralization and carbon peak. For the central and western regions, corresponding industrial strategies should be formulated according to regional characteristics, and it is important to increase infrastructure investment, eliminate backward production capacity, guide rational allocation of resource elements and provide support for industrial upgrading.

Fourth, improve the energy structure and increase the energy utilization rate. Energy structure and energy intensity have a negative impact on upgrading the industrial structure. China’s energy consumption is dominated by coal. By improving energy efficiency and vigorously developing and utilizing clean energy, the energy structure will be effectively improved. This will not only reduce carbon emissions and promote low-carbon development but also contribute to upgrading of the industrial structure.

## Figures and Tables

**Figure 1 ijerph-20-01945-f001:**
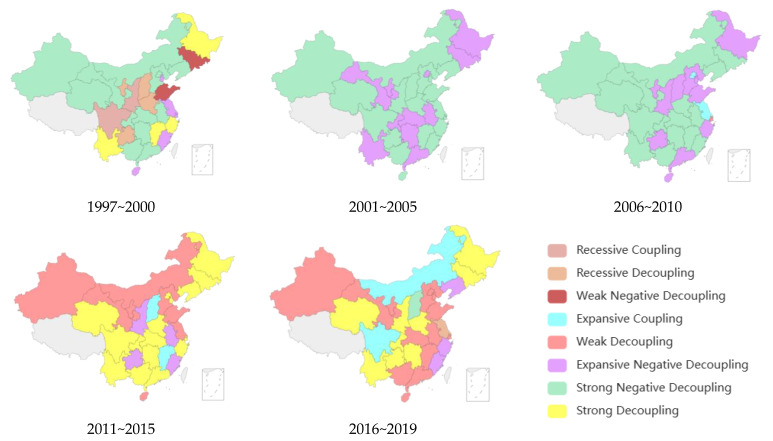
Evolution and distribution diagram of decoupling types.

**Figure 2 ijerph-20-01945-f002:**
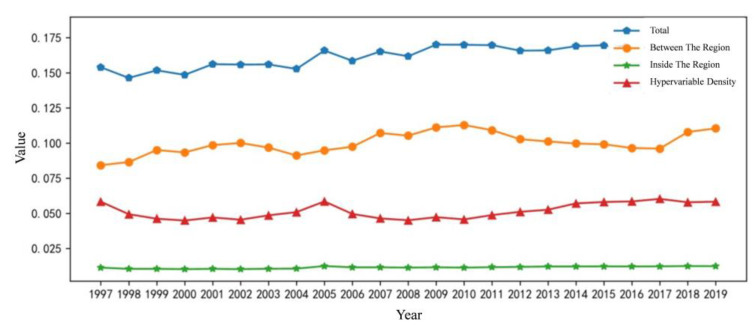
National Dagum Gini coefficient change trend chart.

**Figure 3 ijerph-20-01945-f003:**
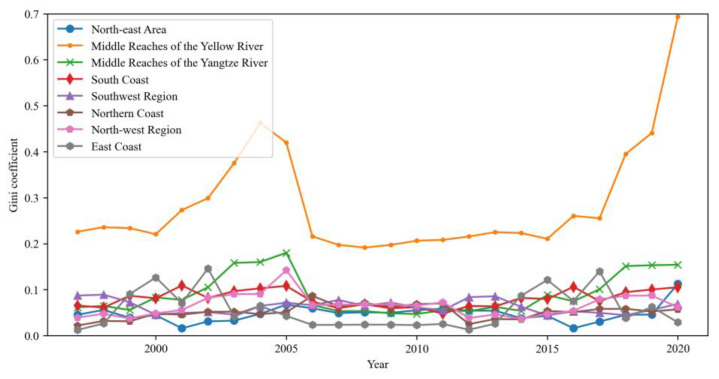
Change trend of Dagum Gini coefficient in different regions.

**Figure 4 ijerph-20-01945-f004:**
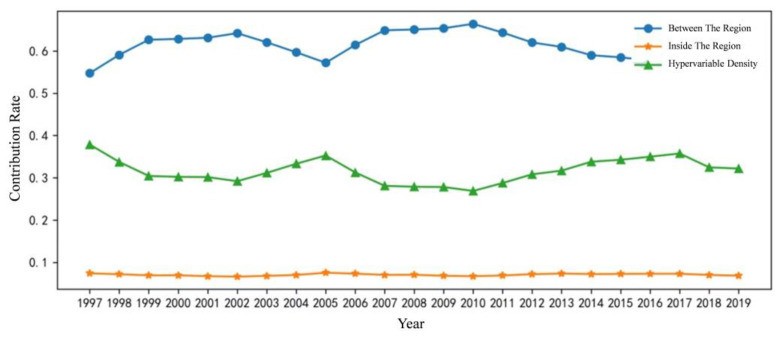
Change chart of national and regional contribution rates.

**Figure 5 ijerph-20-01945-f005:**
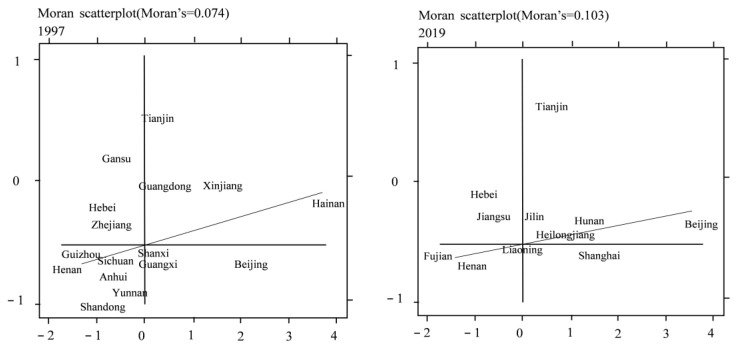
Local Morantu of 1997 and 2019 industrial structure upgrading index.

**Figure 6 ijerph-20-01945-f006:**
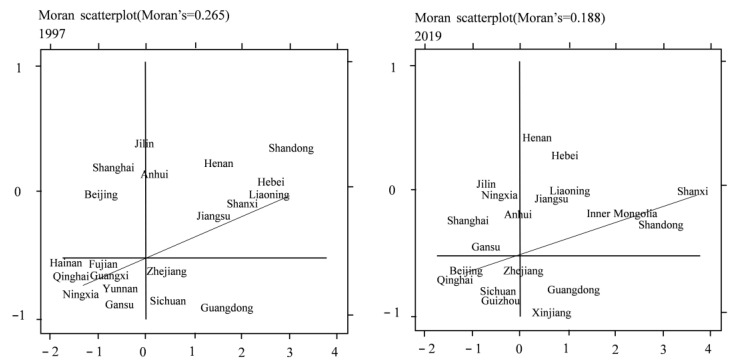
Local Moran charts of carbon emissions in 1997 and 2019.

**Table 1 ijerph-20-01945-t001:** Indicator system.

Indicator Type	First-Level Indicators	Second-Level Indicators	Indicator Code
Interpreted variable	Industrial structure upgrading	Advanced industrial structure	IS
		Rationalization of industrial structure	
Core explanatory variables	Carbon emission	Total carbon dioxide emissions	CI
Intermediary variable	Energy structure	Total coal consumption	ES
	Energy intensity	Proportion of energy consumption	EI
Control variable	Economic development level	Total GDP	AGDP
	Degree of government intervention	Proportion of fiscal expenditure	DI
	Scientific and technological level	R&D investment	RD
	Income level	Gini index	GN

**Table 2 ijerph-20-01945-t002:** Categorization of decoupling status.

Decoupling State		ΔCI/CI	ΔIS/IS	$\varphi_{CI,IS}$
Negative decoupling	Expansive negative decoupling	+	+	(1.2, +∞)
	Strong negative decoupling	+	−	(−∞, 0)
	Weak negative decoupling	−	−	[0, 0.8)
Decoupling	Weak decoupling	+	+	[0, 0.8)
	Strong decoupling	−	+	(−∞, 0)
	Recessive decoupling	−	−	(1.2, +∞)
Coupling	Expansive coupling	+	+	[0.8, 1.2]
	Recessive coupling	−	−	[0.8, 1.2]

**Table 3 ijerph-20-01945-t003:** Division of economic zones.

Economic Region	Province
Southern coast	Fujian, Guangdong, Hainan
Northern coast	Beijing, Tianjin, Hebei, Shandong
Northeast China	Liaoning, Jilin, Heilongjiang
The middle reaches of the Yangtze River	Anhui, Jiangxi, Hubei, Hunan
The middle reaches of the Yellow River	Shanxi, Inner Mongolia, Henan, Shaanxi
Southwest China	Guangxi, Chongqing, Sichuan, Guizhou, Yunnan
East Coast	Shanghai, Jiangsu, Zhejiang
Northwest China	Gansu, Qinghai, Ningxia, Xinjiang

**Table 4 ijerph-20-01945-t004:** Nuclear density estimation results of China and the east, middle and west regions.

	Distribution Position	Shape	Kurtosis
National regions	The overall shift to the right shows that the decoupling index has an upward trend during the review period, and the carbon emission rate is greater than the growth rate of the industrial structure upgrading index	Most years are in a single peak state, indicating that the decoupling index gap between provinces is relatively stable	The peak value reached its maximum in 2013 and then showed a high decline trend, indicating that the distribution difference of decoupling index among provinces first increased and then decreased
Eastern region		In 2019, there was a significant bimodal trend, and the polarized camp has strengthened. The left peak was greater than the right peak, indicating that the number of provinces with low decoupling index was more than with high decoupling index	Before 2013, the nuclear density function peak of carbon emissions was relatively low, with the highest peak in 2013; the peak value decreases, indicating that the difference in decoupling index between eastern regions is enlarged
Central region	The overall shift from left to right shows that the decoupling index first decreases and then increases during the review period, and the carbon emission rate is first lower than and then higher than the growth rate of the industrial structure upgrading index	The density function changes from a bimodal state to a unimodal state. The bimodal state has no significant trend, the camp of polarization changes and the regional polarization weakens	In 2013, there was a changing trend from peak to broad peak, indicating that the difference in decoupling index was expanding
Western region		The density function shows the multi-peak phenomenon for many years, indicating that the carbon emissions in the western region have a greater two-stage differentiation, the decoupling index varies greatly among provinces and the polarization situation is changing	The peak value was the highest in 2013, followed by 2019, indicating widening and strong polarization of the carbon emissions gap in the western region

**Table 5 ijerph-20-01945-t005:** Industrial structure upgrading index and global Moran index of carbon emissions from 1997 to 2019.

Particular Year	IS			CI		
	Moran’s *I*	Z Value	*p* Value	Moran’s *I*	Z Value	*p* Value
1997	0.077	1.063	0.044	0.274	2.535	0.006
1998	0.076	1.011	0.046	0.221	2.114	0.017
1999	0.092	1.222	0.011	0.266	2.501	0.006
2000	0.068	0.994	0.040	0.236	2.261	0.0132
2001	0.075	1.048	0.047	0.266	2.509	0.006
2002	0.084	1.096	0.037	0.278	2.611	0.005
2003	0.059	0.862	0.044	0.275	2.590	0.005
2004	0.055	0.817	0.007	0.279	2.653	0.004
2005	0.030	0.042	0.043	0.297	2.881	0.002
2006	0.040	0.065	0.044	0.271	2.704	0.003
2007	0.024	0.123	0.041	0.249	2.498	0.006
2008	0.023	0.145	0.042	0.259	2.527	0.006
2009	0.020	0.613	0.047	0.283	2.720	0.003
2010	0.058	0.040	0.049	0.301	2.827	0.002
2011	0.050	0.966	0.047	0.309	2.861	0.002
2012	0.043	0.865	0.044	0.289	2.709	0.003
2013	0.038	0.826	0.044	0.276	2.820	0.002
2014	0.021	0.588	0.048	0.250	2.617	0.004
2015	0.014	0.506	0.006	0.254	2.600	0.005
2016	0.036	0.709	0.039	0.226	2.346	0.009
2017	0.050	0.851	0.047	0.242	2.454	0.007
2018	0.089	1.435	0.016	0.205	2.182	0.015
2019	0.107	1.592	0.046	0.195	2.068	0.019

**Table 6 ijerph-20-01945-t006:** LM inspection results table.

Test	LM Value	*p* Value
LM Lag test	2.424	0.120
Robust LM Lag test	31.972	0.000
LM Error test	33.748	0.000
Robust LM Error test	63.296	0.000

**Table 7 ijerph-20-01945-t007:** Regression results of China’s overall model.

Variable Name	SAR	SEM	SDM
lnCI	0.032 ** (2.230)	0.028 ** (2.010)	0.023 * (1.640)
lnES	−0.294 *** (−17.610)	−0.310 *** (−18.740)	−0.300 *** (−17.150)
lnEI	0.260 *** (3.490)	0.318 *** (4.160)	0.268 *** (3.420)
lnAGDP	0.162 *** (11.540)	0.190 *** (12.650)	0.182 *** (7.450)
lnDI	0.016 (1.140)	0.0158 (1.020)	−0.003 (−0.210)
lnRD	−4.625 *** (−4.860)	−5.984 *** (−5.340)	−8.387 *** (1.220)
lnGN	−0.529 ** (−2.430)	−0.634 ** (−2.510)	0.035 (0.120)
Constant	1.471 (12.880)	1.527 (11.530)	1.017 (5.220)
W×lnCI	-	-	0.012 (0.270)
Control variable	-	-	Spatial lag
*Log−L*	485.531	496.183	503.608
$\sigma^{2}$	0.012 ***	0.011 ***	0.011 ***
The goodness of fit	0.526	0.524	0.543
Hausman Test	21.530 **	25.220 **	27.190
*LR* test	36.160 ***	14.850	-

Note: *** *p* < 0.01, ** *p* < 0.05, and * *p* < 0.1.

**Table 8 ijerph-20-01945-t008:** Partial regression results of eastern, central and western models.

		Eastern Region	Central Region	Western Region
SAR	Control variable	-	-	-
	$\sigma^{2}$	0.010 ***	0.005 ***	0.005 ***
	The goodness of fit	0.800	0.506	0.329
	Hausman Test	137.100 ***	455.630 ***	49.230 ***
	*LR* test	25.820 ***	43.250 ***	12.820 *
SEM	Control variable	-	-	-
	$\sigma^{2}$	0.010 ***	0.005 ***	0.005 ***
	The goodness of fit	0.793	0.404	0.315
	Hausman Test	137.820 ***	24.680 **	561.860 ***
	*LR* test	30.010 ***	39.440 ***	10.040
SDM	Control variable	Spatial lag	Spatial lag	Spatial lag
	$\sigma^{2}$	0.009 ***	0.004 ***	0.004 ***
	The goodness of fit	0.817	0.571	0.467
	Hausman Test	-	-	-
	*LR* test	-	-	-

Note: *** *p* < 0.01, ** *p* < 0.05, and * *p* < 0.1.

**Table 9 ijerph-20-01945-t009:** Regression results of east, middle and west models.

Variable Name	Eastern Region	Central Region	Western Region
lnCI	−0.169 *** (−5.810)	0.044 (1.490)	0.003 (0.220)
lnES	−0.309 *** (−14.590)	−0.020 (−0.510)	−0.041 (−1.250)
lnEI	0.136 (1.330)	0.340 *** (1.530)	0.177 (0.860)
lnAGDP	0.469 *** (12.970)	−0.309 *** (−4.120)	−0.255 ** (−2.570)
lnDI	−0.028 (−0.660)	0.145 *** (5.110)	0.002 (0.130)
lnRD	−7.438 *** (−5.010)	−5.202 (−2.250)	3.732 ** (2.170)
lnGN	−1.308 *** (−2.630)	−0.470 (−1.200)	0.826 *** (2.840)
Constant	0.686 ** (2.590)	1.282 *** (3.540)	0.226 (0.830)
W×lnCI	−0.016 (−0.180)	0.083 (1.250)	0.010 (0.260)

Note: *** *p* < 0.01 and ** *p* < 0.05.

**Table 10 ijerph-20-01945-t010:** Decomposition of spatial effects of China as a whole.

Variable Name	Direct Effect	Indirect Effect	Total Effect
lnCI	0.011 (0.840)	−0.107 ** (−2.390)	−0.096 ** (−2.050)
lnES	−0.275 *** (−18.290)	0.128 *** (3.430)	−0.147 *** (−3.730)
lnEI	0.082 (1.220)	−0.523 *** (−3.280)	−0.441 *** (−2.620)
lnAGDP	−0.191 *** (−4.690)	0.005 (0.120)	−0.186 *** (−4.010)
lnDI	−0.088 *** (−5.640)	−0.040 (−1.420)	−0.128 *** (−4.790)
lnRD	−6.989 *** (−6.380)	4.660 * c(1.800)	−2.329 (−0.810)
lnGN	0.198 (0.650)	0.877 * (1.640)	1.075 * (1.750)

Note: *** *p* < 0.01, ** *p* < 0.05, and * *p* < 0.1.

## Data Availability

The data involved in this study are all from public data.

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
