# Peer review of "Study on the Decoupling and Interaction Effect between Industrial Structure Upgrading and Carbon Emissions under Dual Carbon Targets"

_ijerph, 2023, doi:10.3390/ijerph20031945_

Round 1
Reviewer 1 Report
This manuscript systematically identified decoupling and interaction effect between industrial structure upgrading and carbon emissions under dual carbon targets, The topic is meaningful to development of China's economy. However, there are some disadvantages in the manuscript, for example, the lack of substantial innovation has made the contribution of the manuscript limited. The following comments may be helpful for improving this manuscript.
1)In the Introduction, I think the author should elaborate why this research should be carried out and reveal the importance of this research. In addition, the research motivation and innovation of this manuscript need to be strengthened. Now the introduction is not well written. China as developing great power have the corresponding carbon emissions responsibility, therefore, in the introduction part should be stressed that China's per capita carbon emissions level is not high, and Europe and the United States and other developed countries in terms of carbon emissions responsibility should have differences.
2)Literature summary needs refining, should not be a simple list, should through the analysis of existing research, find out the deficiency existing in the existing research and the innovation of this study.
3)Lack the necessary discussion, I think the author should talk more with the existing literature, and explain the content of the status quo analysis as little as possible. The content of the status quo analysis should be placed in the Results.
4)In conclusion, the authors lack of response to the introduction of three questions. That is, What is the decoupling relationship between industrial structure upgrading and carbon emissions under the dual carbon target nationwide? How about the interaction effect? What are the characteristics of space metrology?
5)In this manuscript, the authors state that "policy recommendations to reduce emissions are proposed". But in this manuscript, I cannot feel that detailed and targeted recommendations have been proposed. I think the authors should update the recommendations based on the previous research conclusions.

Reviewer 2 Report
This article makes a detailed and substantial study on China's industrial structure upgrading and carbon emissions. However, there are some deficiencies that need to be further improved, as follows:
1. Avoid syntax, case or tense errors.
-Eg. Line 24. The word "Reveal" should not be capitalized.
2. Line 35-38. The first two sentences of the introduction are exactly the same as the abstract. If possible, briefly introduce the problem of global climate change or describe the process of global carbon neutralization. You can refer to the following paper or the following webpage:
-Zhao Y, Su Q, Li B, et al. 2022. Have those countries declaring “zero carbon” or “carbon neutral” climate goals achieved carbon emissions-economic growth decoupling?. Journal of Cleaner Production, 363: 132450.
-https://eciu.net/netzerotracker
3. The literature review needs to be fully revised.
-Revise the citation format of references under the Author Guidance. (eg. “Ji Yujun et al. (2015)[2]” should be converted into “Ji et al. [2]”.)
-Avoid long and difficult sentences, and use short sentences (eg. Line 62-68).
-The classification of literature is not clear.
Eg.: Research on carbon emission measurement. The author introduced a literature of using the LMDI to research the drivers of carbon emission. It is not appropriate.
Eg.: Research on the relationship between industrial structure upgrading and carbon emission decoupling. In the Ref.[8], can the economic development be regarded as industrial structure upgrading?
Eg.: The carbon emission decoupling model shall be improved, expanded, or combined with other models. The first three items start from the research content, and the last item starts from the improvement or combination of models. This does not seem appropriate.
My personal suggestion is that the literature review can be based on the main research content of the article. For example, how is the current industrial structure upgrading reflected, whether there is a study on the decoupling relationship between industrial structure upgrading and carbon emissions, whether there is a study on industrial structure upgrading and the spatial spillover effect of carbon emissions. The literature review should be closely related to the main contents and innovations of the article. It is necessary to use critical language to summarize the shortcomings of existing literature and the contributions made by this article.
4. What’s the meaning of CI and IS in formula (1)?
5. The name of decoupling state needs further confirmation (eg. Coupling/negative decoupling/decoupling). In addition, “Title 2” is an inappropriate expression in Table 1.
6. The way of regional division is different from the traditional administrative planning. What’s the reason? Besides, it's better to use the form of table, which is clearer.
7. Line 215. How to process the missing values and the outliers. It’s not clear.
8. Formula (17) seems to be a simple weighted average, which has little to do with entropy weight method. Entropy is a thermodynamic concept used to reflect the degree of confusion between indicators.
9. The discussion of the results is not comprehensive. When describing Dagum Gini Coefficients, the author only described the trend of coefficients, but did not explain the reasons for this phenomenon.
10. Please strictly regulate the name of all the figures.
11. For figure 5 and figure 6, the title is XXX in 1997 and 2019, but the results shown in the figure seem to be in 2019?
12. In the spatial spillover effect model, control variables do not seem to be introduced. In addition, the author does not seem to introduce the spatial spillover effect of other control variables? (eg. W*ln ES, W*ln EI, W*ln AGDP…)
13. The conclusion is different form the experiment result. In the first conclusion, the author only repeated the decoupling relationship in each period. There should be more concise summary.
14. The formation of the reference is not right.

Reviewer 3 Report
Dear Authors,
Thank you for the opportunity to read the article titled “Study on the Decoupling and Interaction Effect Between Industrial Structure Upgrading and Carbon Emissions under Dual Carbon Targets”. The paper is well-written and structured, and it attempts to thoroughly examine the relationship between industrial structure upgrading and carbon emissions in China. However, upon carefully reviewing the article, I noticed several shortcomings in the paper that need to be addressed in order for it to be accepted for publication. Please see my comments and suggestions below:
Title
The title maybe revised to “Study on the Synergistic Effect of Industrial Structure and Carbon Emission under the Dual Carbon Targets”. But it is up to the authors.
Abstract
In the Abstract, it is not clear what the research question or objective of the study is. While the issue of climate and the environment is mentioned, it is not clear how the analysis of industrial structure upgrading and carbon emissions aims to address this issue. Furthermore, the authors mention that policy recommendations will be put forward, it is not clear how these recommendations will contribute to the existing literature or address the research question.
Introduction
The background information on the importance of addressing global climate issues and upgrading industrial structure in China is interesting, but it is not clear how these issues are related to the research question or objectives of the study. Similarly, the literature review in the Introduction provides some context on previous research on industrial structure upgrading and carbon emissions, but it is not comprehensive and does not adequately situate the current study within the existing literature. In my opinion, the review could benefit by considering studies from related disciplines such as resource use efficiency and other measures to reduce carbon emissions. This would provide a more comprehensive understanding of the various approaches and strategies that have been implemented or studied in order to address the issue of carbon emissions in China. Additionally, including studies from other countries or regions could provide a valuable comparative perspective and highlight any potential differences or similarities in the relationships between industrial structure upgrading and carbon emissions in different contexts. Here I suggest some studies for the authors to include in the literature review:
https://doi.org/10.3389/fenvs.2022.944156
https://doi.org/10.1504/ijmtm.2021.121110
https://doi.org/10.1007/s11356-022-24736-5
Research Methods
The use of the Tapio index model to construct the "decoupling model of carbon emissions and industrial structure upgrading" is mentioned, but more details are needed, i.e. what this model is or how it is constructed. It would be helpful to provide detail on this model and its assumptions, as well as to explain the relevance of the decoupling concept to the research question.
In addition, the use of the kernel density function and Dagum Gini coefficient as analysis techniques is mentioned. It is pertinent that authors explain how these techniques contribute to the overall research objectives and help to address the research question.
Results
The results presented in Figure 1 suggest that there have been significant changes in the decoupling index and decoupling status of all provinces in China between 1997 and 2019. The authors need to further contexulaized these findings and link them back to the research objectives.
Table 2 presents the results of the kernel density estimation for China and different regions, but it is not clear how these results were obtained or what they mean. Please explain the implications of the different distribution positions, shapes, and kurtosis values for carbon emissions and industrial structure upgrading.
Overall, the analysis in section 3 does not provide sufficient detail or context to fully understand the results or their implications. I recommend revising this section to provide more detail on the methodology used, to explain the meaning of the results, and to provide more context by discussing how these results compare to previous research on the topic. This will help the reader understand the underlying assumptions and logic of the study and how the research question is being addressed.
One potential issue with results of section 4 is the use of the global and local Moran indices to analyze the spatial autocorrelation of the industrial structure upgrading index and carbon emissions. While these indices can provide valuable insights into the spatial patterns of these variables, it is important to consider the limitations and assumptions of these methods. For example, the Moran index assumes that the spatial relationships between observations are constant over the study period, which may not always be the case in real-world situations. Additionally, the use of these indices may not adequately account for other factors that may influence the spatial patterns of the variables being analyzed, such as economic, social, or environmental factors. It would be interesting to know authors response to this.
Conclusion
Finally, the conclusion section lacks the context or comparison to other studies on the relationship between industrial structure upgrading and carbon emissions. While the analysis conducted in this paper may provide some valuable insights, it is important to consider the broader literature on this topic in order to understand the limitations and limitations of the findings presented. Additionally, the conclusion section does not adequately address the potential limitations of the study, such as the reliance on a single study period or the inclusion of only a limited set of variables, which may impact the generalizability of the conclusions.
Round 2
Reviewer 2 Report
Most of my concerns of have been addressed. However, the categorization of decoupling status seems not revised yet. It is very important to maintain the rigor of scientific research. I hope the author will strengthen the reference and reading of authoritative literatures.
Tapio, P. 2005. Towards a theory of decoupling: degrees of decoupling in the EU and the case of road traffic in Finland between 1970 and 2001. Transport Policy, 12(2): 137-151.
Author Response
I have revised the categorization of decoupling status according to the literature. Thank you for your guidance.
Reviewer 3 Report
Authors have revised the manuscript carefully and no further changes are required.
Author Response
Thank you for your guidance and recognition